# Antimicrobial Resistance (AMR) in Italy over the Past Five Years: A Systematic Review

**Marco Montalti** [ID], **Giorgia Soldà, Angelo Capodici \*** [ID], **Zeno Di Valerio** [ID], **Giorgia Gribaudo, Giusy La Fauci** [ID], **Aurelia Salussolia** [ID], **Francesca Scognamiglio** [ID], **Anna Zannoner and Davide Gori** [ID]

Unit of Hygiene, Department of Biomedical and Neuromotor Sciences, Public Health and Medical Statistics, University of Bologna, 40126 Bologna, Italy; marco.montalti7@studio.unibo.it (M.M.); giorgia.solda@studio.unibo.it (G.S.); zeno.divalerio@studio.unibo.it (Z.D.V.); giorgia.gribaudo@studio.unibo.it (G.G.); giusy.lafauci@studio.unibo.it (G.L.F.); aurelia.salussolia@studio.unibo.it (A.S.); frances.scognamigli5@studio.unibo.it (F.S.); anna.zannoner2@studio.unibo.it (A.Z.); davide.gori4@unibo.it (D.G.)

\* Correspondence: angelo.capodici@studio.unibo.it; Tel.: +39-051-209-4802

**Abstract:** Antimicrobial Resistance (AMR) has become a global threat to public health systems around the world in recent decades. In 2017, Italy was placed among the worst-performing nations in Europe by the European Centre for Disease Prevention and Control, due to worryingly high levels of AMR in Italian hospitals and regions. The aim of this systematic review was to investigate the state of the art of research on AMR in Italy over the last five years. The PubMed database was searched to identify studies presenting original data. Forty-three of the 9721 records identified were included. Overall, AMR rates ranged from 3% (in a group of sheep farmers) to 78% (in a hospital setting). The methods used to identify the microorganisms, to test their susceptibility and the criteria adopted for the breakpoint were deficient in 7, 7 and 11 studies, respectively. The main findings of our review were that most studies (79.1%) considered hospitalised patients only, 4 studies (9.3%) analysed non-hospitalised populations only. In addition, only 7 studies were multicentric and no scientific literature on the subject was produced in 7 Italian regions. In order to have a solid basis on the topic for the interventions of public health professionals and other stakeholders, studies analysing the phenomenon should be conducted in a methodologically standardised manner, should include all areas of the country and should also focus on out-of-hospital and community-based care and work settings.

**Keywords:** Antimicrobial Resistance (AMR); Italy; surveillance

## 1. Introduction

Since the discovery of penicillin in 1928, numerous classes of antibiotics have been researched and used to treat patients, revolutionizing healthcare. However, bacteria and other pathogens have continued to evolve so that they can resist the new drugs. In recent years, the rate of new antibiotic discovery has dropped dramatically while the use of antibiotic therapies has steadily increased. It is also for these reasons that Antimicrobial Resistance (AMR) is considered the major upcoming public health threat with an estimated 10 million AMR victims by the end of 2050 [1].

In 2017, the European Centre for Disease Prevention and Control (ECDC) mission report, Italy was placed among the worst faring nations in Europe in this context, due to worryingly high AMR levels in Italian hospitals and regions [2].

Data collected through the European Antimicrobial Resistance Surveillance Network (EARS-Net) estimates that each year more than 670,000 infections occur in the European Union/European Economic Area (EU/EEA) due to bacteria resistant to antibiotics causing approximately 33,000 deaths as a direct consequence of these infections [3]. In the above-

mentioned EU/EEA countries, the estimated cost for health care systems is overall around 1,1 billion euros [4].

If proper blended public health interventions—including antibiotic management programs, the promotion of better hygiene, the use of media campaigns and rapid diagnostic tests—would be implemented, it has been estimated that around 27,000 deaths annually in the EU/EEA could be prevented and around €1.4 billion per year saved [4].

Unfortunately, the ECDC "Surveillance of Antimicrobial Resistance in Europe, 2020 data" showed that during the first year of the COVID-19 pandemic, less engagement in AMR surveillance activities was detected. Besides, it outlines that "in the WHO (World Health Organization) European Region, 20% of countries still reported either having no capacity for generating AMR surveillance data or collecting AMR data only at local level and without a standardized approach" [5].

In 2018, following the adoption of the Global Action Plan on Antimicrobial Resistance (GAP) by the World Health Assembly, which set the goal of having a National Action Plan (NAP) on AMR by 2017, 100 countries had prepared a NAP, and a further 67 had plans in progress [6].

In designing effective strategies to tackle AMR, it is certainly crucial to investigate healthcare workers' knowledge and attitudes towards antibiotic use. In this regard, a study by the ECDC and Public Health England published in 2021 investigated European healthcare workers' attitudes towards antibiotic usage, and the overall finding was an immoderate misuse of antibiotics, with a general increasing trend in antibiotic consumption/prescription [7].

Besides, ECDC reports highlighted several times that there is a north-to-south and west-to-east gradient of resistance, with higher rates observed in the southern and eastern parts of the Region. At the same time, efforts to improve antimicrobial consumption in the region remain uneven, and, as per last WHO and ECDC analyses [8] between 2014 and 2018, there were reductions in total antibacterial consumption only in eight out of from 30 EU/EEA countries of the European Surveillance of Antibiotic Consumption network (ESAC-Net), Italy not being one of them. On the other end, for 2020, an overall decrease in community and hospital sector antibiotic consumption in the EU/EEA was reported by ESAC-Net [9].

Improving awareness and knowledge about AMR should be a collective effort since, as stated in the last Surveillance of Antimicrobial Resistance in Europe 2020 data ECDC REPORT, antimicrobial-resistant bacterial microorganisms cannot be contained within borders or regions and therefore there is a need for concerted action to combat AMR throughout the WHO European Region [5].

In the last year we learned how pandemics can disrupt usual workflows and old habits, especially in medicine, while exposing the weaknesses in national health systems as well as in the deficient collaboration and cooperation systems between countries and continents. In this context we remembered how flexibility, resilience and the constant need for updating one's own knowledge is of the uttermost importance.

At the time of the conception of this study there were no systematic reviews investigating the state of art on AMR in Italy, with this study we aim to analyse how studies were methodologically conducted and to provide a general picture of the phenomenon.

## 2. Results

A total of 9.721 papers were identified by the initial search and 43 were included in the final analysis (Figure 1).

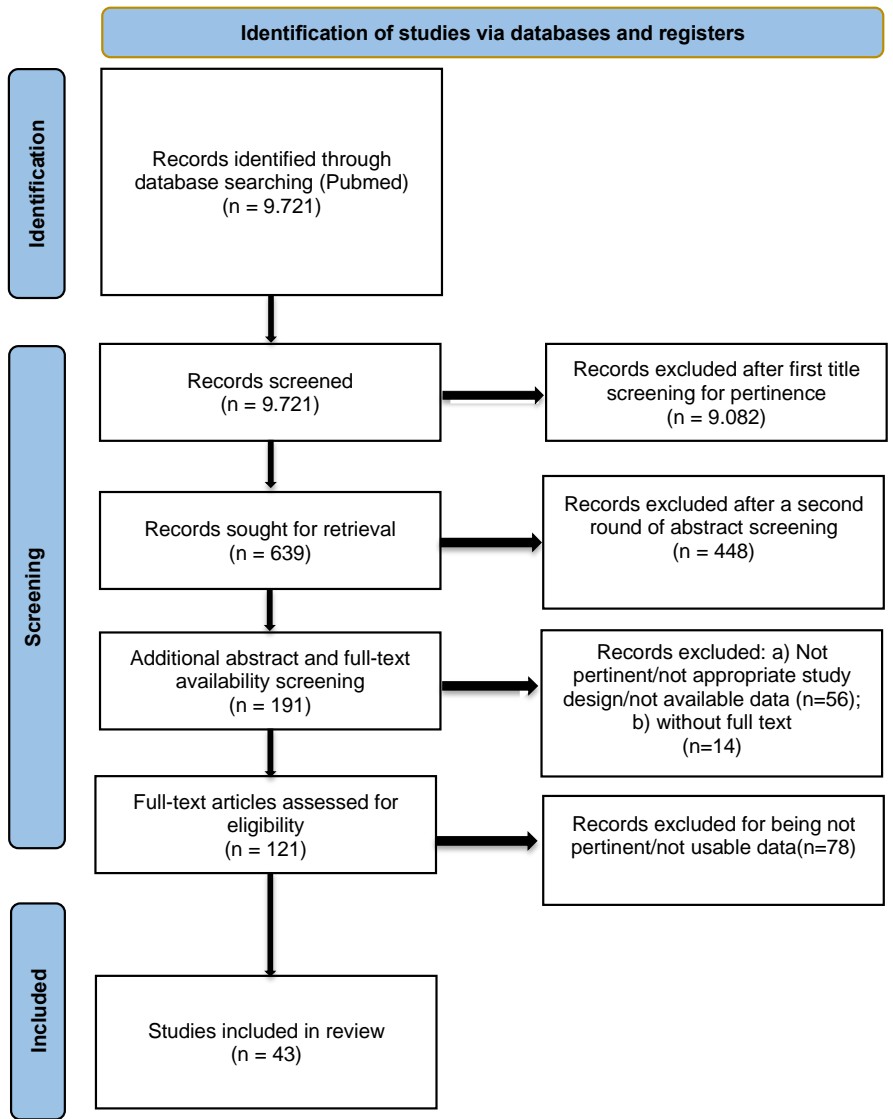

**Figure 1.** Flow chart of the study selection process.

*2.1. Main Characteristics of the Included Studies*

Main sample features of the studies included in the analysis are shown in Table 1. All studies were conducted in Italy and 7 (16.3%) were multicentric. Geographically, the Italian regions in which the largest number of studies included in the review were conducted were Lombardy and Lazio (both *n* = 5, 11.6%), followed by Emilia-Romagna and Campania (*n* = 4, 9.3%). No studies that met the inclusion criteria of the review were found for 7 (16.3%) Italian regions.

**Table 1.** Included studies conducted in Italy focusing on AMR and presenting original data sorted by year of publication.

| Author | Italian Region | Year | Study Design | Study Period | Definition of "Multidrugresistant Isolate" Adopted | Microorganism Studied | Main MDR Microorganism | N° of Isolates | N° of Multidrug Resistant (N, %) | Microorganism Identification Methods | Antimicrobial Susceptibility Testing Method | Breakpoints Used for MIC | Sample Features | Mean Age | Subjects (N) | Setting | Quality Assessment (STROBE) |
|---|---|---|---|---|---|---|---|---|---|---|---|---|---|---|---|---|---|
| Calzi A [10] | Liguria | 2016 | CS | January 2007–December 2014 | n.a. | *E.coli*; Enterobacteriaceae; Paeruginosa | *E. coli* | 3006 | 902 (30%) | Culture-based | BD Phoenix™ Automated Microbiology System | EUCAST | Paediatric patients | 2 | 3364 | Hospital Ward | Intermediate |
| Busani S [11] | Emilia-Romagna | 2016 | CS | January 2008–December 2013 | Non-susceptibility to at least one agent in three or more antimicrobial categories | *Staphylococcus aureus; Enterococcus subspp.;* Enterobacteria ceae; Pseudomonas aeruginosa; and Acinetobacter | *P. aeruginosa* | 115 | 94 (36%) | n.a. | n.a. | n.a. | Adult ICU patients with septic shock | 71 | 381 | Hospital Ward | Poor |
| Bianco A [12] | Calabria | 2016 | LS | Mar 2014–May 2014 | n.a. | *A. baumannii* | *A. baumannii* | 8 | 8 (100%) | MALDI Biotyper® (MBT) | VITEK® 2 system | CLSI | Adult ICU patients | 65 | 8 | Hospital Ward | Good |
| Del Giudice A [13] | Campania | 2016 | LS | 1 January 2008–31 December 2013 | Resistant to at least isoniazid, H, and rifampin, R | *M. tuberculosis* | *M. tuberculosis* | 690 | 31 (4.5%) | GenoType Mycobacterium CM; BD MGIT TBc Identification assay | BD BACTEC™ MGIT™ 960 | n.a. | General population | 42 | 690 | Laboratory of Microbiology and Virology | Good |
| Patriarca F [14] | Friuli-Venezia Giulia | 2016 | CS | January 2013–January 2015 | Non-susceptibility to at least one agent in three or more antimicrobial categories | *K. Pneumoniae; P. aeruginosa; E. coli* | *K. Pneumoniae* | 241 | 13 (5%) | Culture-based | n.a. | EUCAST; CLSI | Adult patients who underwent HSCT | 56 | 241 | Transplant Center | Good |
| Cristina ML [15] | Liguria | 2016 | CS | 2013–2014 | Non-susceptible to imipenem and/or meropenem and/or ertapenem according to the EUCAST breakpoints | *K. pneumoniae* | *K. pneumoniae* | 147 | 147 (100%) | BD Phoenix™ Automated Microbiology System | BD Phoenix™ Automated Microbiology System | EUCAST | Adult patients | 79 | 147 | Hospital Ward | Good |
| Papa V [16] | Sicily | 2016 | CS | May 2014–October 2014 | Non-susceptibility to at least one agent in three or more antimicrobial categories | *S. aureus; S. epidermidis* | CoNS | 131 | 92(33%) | Culture-based | Disk diffusion test | EUCAST | Adult patients | 72 | 120 | Hospital ward | Good |
| Giacobbe DR [17] | Liguria; Piedmont; Emilia-Romagna | 2017 | CS | January 2012–March 2014 | n.a. | *Staphylococcus* spp.; *Enterococcus* spp.; Enterobacteriaceae; non-fermenting Gram negatives; *Candida* spp. | *K. pneumoniae* | 353 | 353 (100%) | MALDI Biotyper® (MBT); VITEK® 2 system | VITEK® 2 system | EUCAST; CLSI | Adult patients | 70 | 353 | Hospital Ward | Intermediate |
| Salerno F [18] | Multicenter | 2017 | LS | January 2007–October 2009 | n.a. | GN; GP | *S. aureus* | 313 | 83 (27%) | n.a. | n.a. | n.a. | Adult patients | n.a. | 203 | Hospital ward | Good |
| Drago L [19] | Lombardy | 2017 | CS | January 2013–June 2015 | n.a. | *Staphylococcus* spp.; Enterobacteriaceae; propionibacterium acnes | *Staphylococcus* spp. | 341 | 144 (42%) | VITEK® 2 system | VITEK® 2 system; E-TEST® strips | n.a. | Adult patients | 65 | 429 | Hospital ward | Good |
| Costa E [20] | Lombardy | 2017 | LS | 2015–2016 | n.a. | *S. aureus*; extended-spectrum b-lactamase; Enterobacterales; Gram-negative bacteria; Enterococci | Extended-spectrum b-lactamase | 577 | 336 (68%) | VITEK® 2 system | VITEK® 2 system | EUCAST | Paediatric patients candidates for cardiac surgery | n.a. | 495 | Hospital ward | Good |
| Proroga YTR [21] | Lazio; Campania | 2017 | LS | January 2013–December 2015 | Magiorakos et al. criteria | *S. enterica* | *S. enterica* | 150 | 90 (60%) | n.a. | Disk diffusion test | CLSI | Adult patients | n.a. | n.a. | Hospital ward | Good |
| Cattaneo C [22] | Multicenter | 2018 | LS | March 2015–August 2015 | Non-susceptibility to at least one agent in three or more antimicrobial categories | VRE; ESBL-P; CarbaR | CarbaR | 2226 | 144 (7%) | Culture-based | Disk diffusion test | EUCAST | Adult patients with a haematological neoplasm | n.a. | 144 | Hospital Ward | Good |
| García-Fernández A [23] | Multicenter | 2018 | LS | January 2013–December 2016 | Non-susceptibility to at least one agent in three or more antimicrobial categories | *Campylobacter* spp. | *C. jejuni* | 176 | 15 (37%) | Culture-based s; multiplex PCR | Disk diffusion test | EUCAST | Paediatric and adult patients | n.a. | 4672 | Enter-Net Italia | Good |

**Table 1.** *Cont.*

| Author | Italian Region | Year | Study Design | Study Period | Definition of "Multidrugresistant Isolate" Adopted | Microorganism Studied | Main MDR Microorganism | N° of Isolates | N° of Multidrug Resistant (N, %) | Microorganism Identification Methods | Antimicrobial Susceptibility Testing Method | Breakpoints Used for MIC | Sample Features | Mean Age | Subjects (N) | Setting | Quality Assessment (STROBE) |
|---|---|---|---|---|---|---|---|---|---|---|---|---|---|---|---|---|---|
| Forcina A [24] | Lombardy | 2018 | LS | July 2012–January 2016 | Non-susceptibility to at least one agent in three or more antimicrobial categories | GNB | *P. aeruginosa* | 54 | 7 (16%) | Culture-based | n.a. | n.a. | Adult patients undergoing autologous and allogeneic transplant | n.a. | 348 | Hospital Ward | Poor |
| Cama BAV [25] | Sicily | 2018 | CS | January 2016–December 2016 | n.a. | *B. melitensis* | *B. melitensis* | 12 | 7 (58%) | Culture-based | n.a. | n.a. | Adult patients | n.a. | 24 | Hospital ward | Good |
| De Angelis G [26] | Lazio | 2018 | CS | 2007–2015 | Non-susceptibility to at least one agent in three or more antimicrobial categories | *E. coli; E. faecium; S. aureus; K. pneumoniae; A. baumannii; P. aeruginosa; Enterobacter* spp. | *E. coli* | 9720 | 5336 (54.9%) | VITEK 2® system; MALDI Biotyper® (MBT) | VITEK® 2 system; MERLIN Diagnostica GmbH | EUCAST | General population | n.a. | n.a. | Laboratory of Microbiology and Virology | Good |
| Mascaro V [27] | Calabria | 2019 | CS | March 2017–February 2018 | Non-susceptibility to at least one agent in three or more antimicrobial categories | *S. aureus* | *S. aureus* | 95 | 3 (3%) | Gram stain, catalase, and coagulase tests(Pastorextm Staph-plus Bio-Rad), API Staph identification system (bioMérieux) | Disk diffusion test | EUCAST | Sheep farm workers | 46 | 275 | Farm | Poor |
| Loconsole D [28] | Apulia | 2019 | CS | January 2013–April 2015 | n.a. | Macrolide Resistant *M. pneumoniae* | Macrolide Resistant *M. pneumoniae* | 15 | 3 (34%) | RT-PCR | RT-PCR; MLVA | n.a. | Adult patients | 53 | 234 | Hospital ward | Good |
| Del Prete R [29] | Apulia | 2019 | CS | January 2015–December 2017 | Non-susceptibility to at least one agent in three or more antimicrobial categories | *K. pneumoniae* | *K. pneumoniae* | 439 | 439 (58%) | VITEK® 2 system | VITEK® 2 system | EUCAST | Adult patients | n.a. | 356 | Hospital ward | Good |
| La Fauci V [30] | Sicily | 2019 | LS | June 2017–May 2018 | Magiorakos et al. criteria | Staphylococcus; Enterobacteria; Pseudomonas; Acinetobacter; Rhizobium; Sphingomonas; Ochrobactrum; *Streptococcus* spp.; Aerococci; Burkholderia; Roseomonas; Kytococcus. | *Staphylococcus* spp. | 608 | 47 (15%) | VITEK® 2 system | VITEK® 2 system | EUCAST | Adult patients | n.a. | n.a. | Hospital Ward | Poor |
| Mascaro V [31] | Calabria | 2019 | CS | May 2017–March 2018 | Non-susceptibility to at least one agent in three or more antimicrobial categories | *S. aureus* | *S. aureus* | 101 | 10 (10%) | API Staph identification system (bioMérieux) | Disk diffusion test | EUCAST | Athletes | 23 | 238 | Public or private gyms | intermediate |
| Grandi G [32] | Piedmont | 2019 | LS | 1988–2017 | Non-susceptibility to at least one agent in three or more antimicrobial categories | *S. aureus; Staphylococcus spp; S. pneumoniae; P. aeruginosa; H. influenzae; Streptococcus* spp. | *S. aureus* | 2898 | n.a. (8.7%); n.a. (10%); 348 (12%) | n.a. | Disk diffusion test | EUCAST; CLSI | Adult patients with ocular infection | n.a. | n.a. | Hospital Ward | Good |
| Pirolo M [33] | Calabria | 2019 | CS | March 2018–February 2019 | Non-susceptible to at least three non β-lactams antimicrobial classes | *S. aureus* | *S. aureus* | 49 | 19 (9%) | Staphytect plus test; PCR | VITEK® 2 system | CLSI | Pig farm workers | 46 | 88 | Farm | Intermediate |
| Cannas A [34] | Lazio | 2019 | CS | 2011–2016 | Resistance to isoniazid and rifampicin | *M. tuberculosis* | *M. tuberculosis* | 926 | 51 (6%) | Ziehl Nielsen; hot staining; mRNA testing (E-MTD, TRCReady-80) | Proportion dilution | n.a. | Adult patients | 40 | 926 | Hospital Ward | Intermediate |
| Tumbarello M [35] | Lazio; Lombardy | 2020 | CS | 1 January 2016–31 December 2017 | Non-susceptibility to at least one agent in three or more antimicrobial categories | *P. aeruginosa* | *P. aeruginosa* | 242 | 65 (27%) | MALDI Biotyper® (MBT) | VITEK® 2 system; MERLIN Diagnostica GmbH | EUCAST | Adult patients | 71 | 305 | Hospital Ward | Good |

**Table 1.** *Cont.*

| Author | Italian Region | Year | Study Design | Study Period | Definition of "Multidrugresistant Isolate" Adopted | Microorganism Studied | Main MDR Microorgan-ism | N° of Isolates | N° of Multidrug Resistant (N, %) | Microorganism Identification Methods | Antimicrobial Susceptibility Testing Method | Breakpoints Used for MIC | Sample Features | Mean Age | Subjects (N) | Setting | Quality Assessment (STROBE) |
|---|---|---|---|---|---|---|---|---|---|---|---|---|---|---|---|---|---|
| Papalini C [36] | Umbria | 2020 | CS | 2014–2019 | Magiorakos et al. criteria | *K. pneumoniae* | *K. pneumoniae* | 3 | 3 (100%) | MALDI Biotyper® (MBT) | BD Phoenix™ Automated Microbiology System | EUCAST | General population | n.a. | 113 | Laboratory of Microbiology and Virology | Intermediate |
| Riccardi N [37] | Lombardy | 2020 | CS | 1 January 2000–1 Jan 2015 | n.a. | *M. tuberculosis* | *M. tuberculosis* | 8603 | 370 (4%) | n.a. | n.a. | n.a. | Adult migrant patients | 32 | 116 | Hospital Ward | Intermediate |
| Pompilio A [38] | Lazio | 2020 | CS | 2017–2018 | Non-susceptibility to at least one agent in three or more antimicrobial categories | *S. maltophilia* | *S. maltophilia* | 85 | 66 (78%) | Thermo Scientific™ Culti-Loops™ API 20NE; VITEK® 2 system | Disk diffusion test; broth microdilution method | CLSI | Pediatric patients | n.a. | n.a. | Hospital Ward | Good |
| Seminari E [39] | Lombardy | 2020 | LS | 1 January 1998–31 December 2017 | Resistance to isoniazid and rifampicin | *M. tuberculosis* | *M. tuberculosis* | 919 | 28 (3%) | n.a. | Culture based identification methods; Mycobacteria Growth Indicator Tube (MGIT) | n.a. | Adult patients | 47 | 919 | Hospital Ward | Intermediate |
| Loconsole D [40] | Apulia | 2020 | CS | 2014–2016 | n.a. | *K. pneumoniae* | *K. pneumoniae* | 691 | 691 (100%) | Cepheid's GeneXpert® System | n.a. | EUCAST | Adult patients | n.a. | 691 | Hospital ward | Good |
| Gudiol C [41] | Italy | 2020 | CS | 1 January 2006–31 May 2018 | Non-susceptibility to at least one agent in three or more antimicrobial categories | *P. aeruginosa* | *P. aeruginosa* | 123 | 50 (41%) | Culture-based identification methods | n.a. | EUCAST; CLSI | Adult neutropenic onco-hematological patients | n.a. | 123 | Hospital Ward | Good |
| Fiorini G [42] | Emilia-Romagna | 2020 | CS | 2009–2019 | n.a. | *H. pylori* | *H. pylori* | 294 | 294 (100%) | Culture-based | E-TEST® strips | EUCAST | Adult migrant patients | 41 | 294 | Hospital ward | Good |
| Gentile B [43] | Emilia-Romagna | 2020 | CS | 2013–2014 | n.a. | CR-*K. pneumoniae* | CR-*K. pneumoniae* | 27 | 27 (100%) | Illumina MiSeq platform | VITEK® 2 system | EUCAST | Adult patients | 72 | 26 | Hospital ward | Good |
| Saracino IM [44] | Emilia-Romagna | 2020 | CS | 2016–2019 | n.a. | *H. pylori* | *H. pylori* | 663 | 33% | Culture-based | E-TEST® strips | EUCAST | Adult patients | 51 | 270 | Hospital ward | Intermediate |
| Normanno G [45] | Veneto | 2020 | CS | 2017–2018 | n.a. | MRSA | MRSA | 4 | 4 (100%) | Disk diffusion test | Disk diffusion test | CLSI | Cow farm workers | n.a. | 24 | Farm | Intermediate |
| Mascellino MT [46] | Lazio | 2020 | CS | 2017 | Resistance to more than one antibiotic | *H. pylori* | *H. pylori* | 80 | a) 24 (30%); b) 11 (14%); c) 9 (11%); d) 5 (6%) | Pylori Agar; GenoType® HelicoDR test | E-TEST® strips | EUCAST | Adult patients | 59 | 80 | Hospital ward | Good |
| Karruli A [47] | Campania | 2021 | LS | 9 March 2020–1 May 2020 | Magiorakos et al. criteria | *K. pneumoniae; A. baumannii; P. aeruginosa; Enterobacter* spp.; *S. maltophilia; Enterococcus* spp.; *E. faecium; S. aureus* | *K. pneumoniae* | 32 | 16 (50%) | n.a. | Thermo Scientific™ Sensititre™ | n.a. | Adults ICU patients with SARS-CoV-2 infection | 68 | 32 | Hospital ward | Intermediate |
| Barbadoro P [48] | Marche | 2021 | LS | February 2018–September 2018 | Magiorakos et al. criteria | *K. pneumoniae; E. coli* | *K. pneumoniae* | 2478 | 21 (1%) | VITEK® 2 system | SensiQuattro Gram-negative System | EUCAST | Adult patients | n.a. | 2478 | Hospital ward | Intermediate |
| Gasperini B [49] | Marche | 2021 | CS | (a) Dec 2019Feb 2020; (b) May 2020–July 2020 | Non-susceptibility to at least one agent in three or more antimicrobial categories | *E. coli; Klebsiella* spp.; *Enterococcus* spp.; *Proteus* spp.; *Pseudomonas* spp.; *Enterobacter* spp.; *Staphylococcus* spp. | *E. coli* | a) 36; b) 47 | a) 18 (50%); b) 28 (59.6%) | Culture-based | n.a. | EUCAST | Adult patients | a) 89; b) 86 | a) 33; b) 40 | Hospital ward | Good |
| Magi C [50] | Marche | 2021 | CS | October 2018–May 2019 | n.a. | *K. pneumoniae* | *K. pneumoniae* | 650 | 18 (3%) | MALDI Biotyper® (MBT); VITEK® 2 system | Brilliance™ CRE Agar | EUCAST | General population | n.a. | n.a. | Laboratory of Microbiology and Virology | Intermediate |

**Table 1.** *Cont.*

| Author | Italian Region | Year | Study Design | Study Period | Definition of "Multidrug-resistant Isolate" Adopted | Microorganism Studied | Main MDR Microorganism | N° of Isolates | N° of Multidrug Resistant (N, %) | Microorganism Identification Methods | Antimicrobial Susceptibility Testing Method | Breakpoints Used for MIC | Sample Features | Mean Age | Subjects (N) | Setting | Quality Assessment (STROBE) |
|---|---|---|---|---|---|---|---|---|---|---|---|---|---|---|---|---|---|
| Posteraro B [51] | Lazio | 2021 | CS | 1 March 2020–31 May 2020 | Non-susceptibility to at least one agent in three or more antimicrobial categories | *S. aureus; Enterobacter spp.; E. faecalis; Candida spp.; P. aeruginosa* | *S. aureus* | 69 | 27 (39%) | MALDI Biotyper® (MBT) | VITEK® 2 system; Sensititre YeastOne | EUCAST; CLSI | Adults patients with SARS-CoV-2 infection | 70 | 46 | Hospital ward | Good |
| Petrillo F [52] | Campania | 2021 | CS | 2017–2020 | n.a. | *S. aureus;* Coagulase-negative staphylococci | Coagulase negative staphylococci | 322 | 96 (61%) | Culture-based | MicroScan WalkAway 96 Plus | EUCAST | Adult patients | n.a. | 322 | Hospital ward | Intermediate |

All studies were published between 2016 and 2021 [10–52]. However, the studies examined very heterogeneous periods of time between 1988 [32] and 2020 [47–49,51,52]. The study that analysed the longest time period described the AMR found between 1988 and 2017 [32], the one with the shortest time period between March and May 2020 [47]. Among the studies included in our review, 30 (70%) were cross-sectional, and 13 (30%) longitudinal.

A sensitivity analysis was performed in a representative sample of the studies selected in order to test heterogeneity in the groups using $I^2$ Statistics. Due to the high heterogeneity that emerged (>50%), the meta-analysis was not conducted.

## 2.2. Multidrug-Resistant Microorganisms

Overall, the studies included in the review analysed more than 30 different microorganisms. In the study by La Fauci et al. [30] as many as twelve different microorganisms were taken into account, whereas in most studies (*n* = 23, 53.5%) only one microorganism was analysed [12,13,15,21,23,25,27,29,31,33–42,44–46,50].

The prevailing definition of 'multidrug-resistant microorganism (MDR)' adopted was that of non-susceptibility to at least one agent in three or more antimicrobial categories (used in 17 studies, 39.5%). Sixteen studies (37.2%) did not give a definition of multidrug resistant isolate, while five (11.6%) specifically mentioned the criteria of Magiorakos et al. [53].

The main MDR was found to be *K. pneumoniae* in 9 (20.9%) studies, followed by *S. aureus* (5, 11.6%) and by *P. aeruginosa* and *M. tuberculosis* (both *n* = 4, 9.3%).

The number of isolates collected and analysed within each study ranged from 3 [36] to 3006 [10]. The percentage of MDR detected in relation to the total number of samples analysed ranged from 3%, found by Mascaro et al. in a group of sheep farmers [27], to 78%, found by Pompilio et al. in a hospital environment [38], excluding studies that only analysed MDR which therefore found a resistance rate of 100% (for *K. Pneumoniae* [15,17,36,40,43], *A. baumanii* [12], *H. pylori* [42], *S. aureus* [45]).

In terms of the methods used for the identification of microorganisms, 9 (20.9%) studies used culture-based identification methods as well as other 9 (20.9%) studies used the VITEK 2 system (bio-Mérieux, Marcy l'Etoile, France). Seven (16.3%) studies did not show which method was used to identify the microorganisms.

To test the susceptibility of microorganisms, 11 (25.6%) studies used the VITEK 2 system (bio-Mérieux, Marcy l'Etoile, France), which was the most frequently used method, 9 (20.9%) studies used the disk diffusion test, while 7 (16.3%) studies did not mention any method.

Finally, 22 (51.2%) of the studies included in the review adopted breaking points recommended by the European Committee on Antimicrobial Susceptibility Testing (EU-CAST) [54], 5 (11.6%) studies adopted the Clinical and Laboratory Standards Institute (CLSI) criteria [55], 5 (11.6%) studies adopted both the two most commonly used systems worldwide (EUCAST and CLSI) and 11 (25.6%) studies did not mention any specific criteria.

## 2.3. Samples and Settings

The human sample sizes of studies included in the review were very heterogeneous: from *n* = 8 [12] to *n* = 4672 [23]. In six (14%) studies the sample size studied was not reported. The mean age ranged from 32 [37] to 89 [49] for the samples considering an adult population, 19 (44.2%) studies did not report the demographic characteristics of their study sample, and only one study that considered the paediatric population reported the mean age of the children from whom the samples were taken (with a mean age of 2 years [10]).

The majority of the studies focused on adults and the general population (*n* = 26, 60.5%), six (14%) studies focused on adults with specific pathological conditions, 4 (9.3%) on paediatric populations, 3 (7%) on farm workers (cows, sheep and pigs), 2 (4.7%) on migrant populations and only one study (2.3%) on athletes.

Most of the studies (*n* = 34, 79.1%) considered hospitalised patients, and the hospital ward was the setting in which most of the isolates were collected. Five studies (11.6%) used data from samples analysed by microbiology laboratories, thus including both hospi-

talised and non-hospitalised populations. Only 4 studies (9.3%) analysed non-hospitalised populations only, by sampling on farms [27,33,45] or in public/private gyms [31].

*2.4. Quality Assessment*

Following the descriptive analysis, we assessed the quality of each study. According to the "Strengthening the Reporting of Observational Studies in Epidemiology" (STROBE), all the 43 studies classified as observational had a quality level from Poor to Good: 4 (9.3%) studies were of Poor Quality, 14 (32.6%) studies of Intermediate Quality and 25 (58.1%) studies of Good Quality (Table 1).

## 3. Discussion

Antimicrobial Resistance (AMR) has become one of the most serious threats to public health, accelerated both by the overuse and misuse of antimicrobials in humans and animals and by inadequate infection prevention measures [56].

Among the measures the scientific community has in place to respond to the challenges posed by AMR, in addition to reducing the use of antimicrobials, developing new antimicrobials, improving knowledge of the ecology of resistant bacteria and resistant genes, and increasing stakeholder awareness of the prudent use of antibiotics, the strengthening of the AMR surveillance system in human and animal populations must be considered [57]. Surveillance and epidemiological studies are key tools to prevent the consequences of AMR on public health and the environment.

The AMR situation in Italy has been repeatedly described as worse than in many other European countries and the local data presented during the ECDC Country Visit to Italy also confirmed this trend [2]. As the Country Visit experts pointed out, both robust scientific research in this field and surveillance activities are needed to address the problem. This systematic review provides a snapshot of the current state of research in this field and an understanding of which areas should be further enhanced.

We found that over the past five years only 43 studies on MDR in humans and presenting original data were published in Italy. Excluding studies that focused solely on MDROs (Multi Drug Resistant Organism), resistance rates found in the isolates analysed were ranging between 3%, found by Mascaro et al. in a group of sheep farmers [27], and 78%, found by Pompilio et al. in a hospital environment [38].

Only a few studies were multicentric and in some Italian regions no scientific literature was produced on the topic. AMR is indeed a transregional challenge, especially since the movement of humans (and resistant bacteria) is not limited by regional borders. In a regional framework such as Italy's, it would be necessary to address the issue of AMR with an action that is as cohesive and standardised as possible.

The quality of the studies analysed according to STROBE, was good/intermediate (with 58% of the papers being of good quality). However, critical issues related to the methodology were present, such as the lack of a reliable, standardized method for microorganisms identification and susceptibility testing. Furthermore, various authors did not describe what criteria they used for breakpoints, therefore invalidating their scientific contribution, having seen that their studies were not reproducible. This creates a gap in knowledge that cannot be ignored, since public health professionals can't take policy decisions without rigorous scientific support. Again, for some of the studies included in the review, the demographic characteristics of the human sample taken into consideration, such as mean age, were not present. Knowing the demographic as well as the social characteristics of people with MDR infections could be of great importance to enable AMR and public health professionals to target awareness campaigns on the issue, as recommended by the ECDC [2], and to make antimicrobial stewardship actions more effective [58].

Finally, it should be noted that the vast majority of sample collection settings were hospitals (wards, intensive care units or a transplant centre) while only a small number of studies were conducted in the community (farmers, athletes, etc.). The issue of AMR is

certainly more perceived at the hospital level where the means to do research on the subject are also easier to find. However, the scarcity of data on the non-hospitalised population may reflect the little sense of urgency about the current AMR situation and a tendency of many stakeholders to avoid taking charge of the problem, as already highlighted by ECDC experts [2].

### 3.1. Recommendations

To analyse such a complex topic as AMR, other steps than surveillance should be taken into account such as enhancing infrastructure, conducting antimicrobial steward-ship campaigns, and increasing multidisciplinarity. However, as highlighted in our study, surveillance studies analysing AMR should be conducted in a methodologically standard-ised way and with a global view of the phenomenon. Not only is it necessary to study the phenomenon in areas of the country where little data are available, but it is also essential to bring the focus of public health professionals and other stakeholders to the community and out-of-hospital care and work settings.

### 3.2. Limitations and Strengths

Our study has a few limitations; first, only one database (PubMed) was searched. Furthermore, it is possible that some terms that deserved to be added to the search string were overlooked. Finally, as some authors may prefer their mother tongue and only English-language studies were included in the review, we may have overlooked some studies. Despite these limitations, with this systematic review we managed to fill the gap pointed out by the ECDC during its country visit in 2017 delineating the current research on AMR in Italy with a methodological point of view.

### 4. Materials and Methods

We conducted a systematic review, following the Preferred Reporting Items for Sys-tematic Reviews (PRISMA) approach [59], although the study protocol was not registered. The initial search was implemented on July 8, 2021. The search query consisted of terms considered pertinent by the authors. We searched for publications on Pubmed using the following search string: "((Italy OR Italian) AND ((antibio\*) OR (antimicr\*) OR (drug) AND (resist\*))) AND (("2016/07/08"[Date-Entry]: "2021/07/08"[Date-Entry]))". We included full-text accessible English-language articles. We excluded studies that did not focus on humans, on the Italian territory, and studies which did not focus on antibiotic resistance. The Prisma Flowchart is displayed in Figure 1.

### 4.1. Data Extraction

Data was extracted by seven independent reviewers (MM, GS, AC, ZDV, GG, GLF and AZ). Disagreement on extracted data was discussed with an independent arbiter (DG). On the basis of the title and abstract, eligibility for the article was determined, and the full text of the selected papers provided information for the final decision of inclusion or exclusion. A manual search was also performed by reviewing bibliographies of pertaining articles to identify additional studies.

Descriptive variables extracted from each article were: "Title", "Author", "Italian Region", "Publication Month/Year", "Study Design", "Study Period", "Definition of MDR isolate adopted", "Microorganism Studied", Main MDR Microorganism", "N° of isolates", "MDR %", "Microorganism Identification Methods", "Antimicrobial Susceptibility Testing Method", "Breakpoints used for MIC", "Sample Features", "Mean age", "N° of subjects" and "Setting".

### 4.2. Quality Assessment

Eight authors (MM, AC, ZDV, GG, GLF, AS, FS and AZ) independently and blindly assessed the quality of the included studies using the "Strengthening the Reporting of Observational Studies in Epidemiology (STROBE) tool for observational studies" [60]. Any

disagreement between the researchers was resolved through discussion. If discussion was not sufficient a blind reviewer (GS) was appealed as a tiebreaker. The STROBE statement is a 22-item tool specifically designed to evaluate observational studies quality. 18 items are the same in the three different checklists and five questions (6-12-14-15) are differently formulated for each study design: (1) Cohort study, (2) Case report study, (3) Cross-sectional study. STROBE does not provide ways to clearly define a score allowing to rate the quality of the study. As a general rule, the higher the score, the higher the quality of the study. We decided to use the cut-offs for three levels of score: 0–14 as poor quality, 15–25 as intermediate quality and 26–33 as good quality of the study.

**Author Contributions:** Conceptualization and methodology, all authors contributed equally; formal analysis and investigation, M.M., G.S., A.C., Z.D.V., G.G., G.L.F., A.S., F.S. and A.Z.; writing—original draft preparation, M.M., G.S. and A.C.; writing—review and editing, all authors contributed equally; supervision, M.M. and D.G. All authors have read and agreed to the published version of the manuscript.

**Funding:** This research received no external funding.

**Institutional Review Board Statement:** Not applicable.

**Informed Consent Statement:** Not applicable.

**Data Availability Statement:** All data will be made available upon reasonable request.

**Conflicts of Interest:** The authors declare no conflict of interest.

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
