# Peer review of "Antimicrobial Resistance (AMR) in Italy over the Past Five Years: A Systematic Review"

_biologics, doi:10.3390/biologics2020012_

Round 1

Reviewer 1 Report

The present study is a comprehensive analysis of the burden of AMR for last 5 years in Italy. Authors attempt to provides a current scenario of research in this field and an understanding of which areas should be further enhanced. This could be an interesting study once the methodology, result sections and conclusions are improved. I feel this unique dataset has not been utilized to its full extent, a more-in depth analysis/interpretation of the data is needed. Many of the relevant studies published recently are missing. [PLoS ONE 17(3): e0265010; Antimicrobial Resistance & Infection Control volume 11, Article number: 45 (2022) ] Overall manuscript is written as such it is difficult to follow and very confusing. I would like to suggest the authors to follow a similar previous study for structuring the manuscript (The LANCET 399(10325): P629-655, 2022; Antimicrobial Resistance & Infection Control volume 11, Article number: 45 (2022).

Abstract is missing important findings.

Please explain the criteria’s for sensitivity testing using I2 statistic, it’s not clear how did they use this information.

The results are not clearly explained, interpretation of the salient findings is missing. I would suggest the author to leverage upon tables and figures to show their data. Please use a separate table for sub heading Multidrug-resistant microorganisms.

The current version needs a major overhaul prior to be publication ready.

Author Response

  • I feel this unique dataset has not been utilized to its full extent, a more-in depth analysis/interpretation of the data is needed. --> We feel much has been done with said database, but if the reviewer has specific ideas we could try and implement them.
  • Many of the relevant studies published recently are missing. [PLoS ONE 17(3): e0265010; Antimicrobial Resistance & Infection Control volume 11, Article number: 45 (2022) ]--> Many studies had to be excluded since our inclusion/exclusion criteria were very strict in order to be as precise as possible with our review's aim. We decided to include "full-text accessible English-language articles. We excluded studies that did not focus on humans, on the Italian territory, and studies which did not focus on antibiotic resistance". With this last sentence we intended to include studies that focused on the study of clinically or laboratoristically evident resistances, therefore studies such as the one brought to our attention by the valuable reviewer's comment did not fall into our inclusion criteria.
  • Overall manuscript is written as such it is difficult to follow and very confusing.I would like to suggest the authors to follow a similar previous study for structuring the manuscript (The LANCET 399(10325): P629-655, 2022; Antimicrobial Resistance & Infection Control volume 11, Article number: 45 (2022). --> We thank the reviewer for his/her valuable inputs. We have added a few sub-paragraphs.
  • Abstract is missing important findings --> We underlined what our main findings were and added one other main finding.
  • Please explain the criteria’s for sensitivity testing using I2 statistic, it’s not clear how did they use this information. --> According to Cochrane the use of I2 statistics can be useful to determine whenever considered studies present excessive variability to potentially conduct a meta-analysis. Even though the I2 statistic is not an absolute value to determine if a meta-analysis is possible, when the number of studies included in computing is reasonably high the reliability of said statistic is reasonably sound.
  • The results are not clearly explained, interpretation of the salient findings is missing. I would suggest the author to leverage upon tables and figures to show their data. Please use a separate table for sub heading Multidrug-resistant microorganisms. --> We thank the reviewer for his/her important comments, nonetheless our aim was to address how studies on MDR were methodologically conducted in the last 5 years in italy. We feel our aim was not clearly stated since this doubts arised in the reviewer's mind, therefore we better clarified our aim.

Reviewer 2 Report

The paper entitled “Antimicrobial resistance (AMR) in Italy over the past five years: a systematic review” by Montalti et al. addresses an interesting subject. Moreover, the methodology and the presentation are correct. However, the manuscript requires a minor revision before publication.

All the reviewer's comments are included throughout the attached document because of the absence of the line numbering.

Author Response

We would like to appreciate the Reviewer feedback on our manuscript and we are thankful for the opportunity given to address their suggestions and concerns. We believe most of their suggestions were incorporated into our revision.
Again, we thank the Reviewer for their time and effort to review this manuscript, and hope they find the revised version satisfactory. All the revisions made to the manuscript are tracked.

We thank the reviewer for his/her precious comments. We have modified the manuscript accordingly to its every comment.

Reviewer 3 Report

The present review entitled “Antimicrobial resistance (AMR) in Italy over the past five years: a systematic review” is an overview of the research on AMR in Italy over the last five years. 

Minor revision 

Introduction should be more focused on the AMR history and present scenario of Italy.  

Was there was any risk of bias with high representation and low representation?

0.5 million cases of N. gonorrhoeae developed in 2013 in Italy, there has been no mention of this study by WHO.

Authors should throw some light to the limitations and the strength of the study as well.

The discussion part is very brief. It would be nice if the authors could split up the discussion into sections like Key findings, and recommendations. The authors focus on surveillance as the key tool however, there are many more factors such as need for development like enhanced infrastructure for research, and epidemiology studies.

Please italicize the names of microbes throughout the manuscript! The font and font size are not consistent throughout the manuscript.

Author Response

  • Introduction should be more focused on the AMR history and present scenario of Italy.  --> We thank the reviewer for his/her precious comments. We added a short paragraph on AMR history. Regarding the present scenario of Italy we sadly cannot add more, since the AMR status has not been properly assessed as also stated by the 2017 ECDC report, and with this paper we aimed to fill this gap, and give directions for future studies to better analyze the AMR status in Italy.
  • Was there was any risk of bias with high representation and low representation? --> We assessed every study that fall into our selection criteria therefore we do not believe there was any representation bias.
  • 0.5 million cases of N. gonorrhoeae developed in 2013 in Italy, there has been no mention of this study by WHO. --> Our review focused its efforts to the last five years, starting 2021. Therefore, albeit important, that study was not included.
  • Authors should throw some light to the limitations and the strength of the study as well. --> We added a paragraph regarding limitations and  strengths
  • The discussion part is very brief. It would be nice if the authors could split up the discussion into sections like Key findings, and recommendations. The authors focus on surveillance as the key tool however, there are many more factors such as need for development like enhanced infrastructure for research, and epidemiology studies. --> We assessed this concern in the manuscript.
  • Please italicize the names of microbes throughout the manuscript! The font and font size are not consistent throughout the manuscript. --> These concerns were addressed throughout the entire manuscript.

Round 2

Reviewer 1 Report

In my opinion, the revised version of the manuscript can be conditionally accepted for publication on “Biologics”. I believe the authors handled the most important issues that came up in the previous round very well. The manuscript may be accepted.